# Video Compressive Sensing Reconstruction Using Unfolded LSTM

**DOI:** 10.3390/s22197172

**Published:** 2022-09-21

**Authors:** Kaiguo Xia, Zhisong Pan, Pengqiang Mao

**Affiliations:** 1College of Communication Engineering, Army Engineering University of PLA, Nanjing 210001, China; 2College of Command and Control, Army Engineering University of PLA, Nanjing 210001, China

**Keywords:** video compressing sensing, end-to-end deep learning network, unfolded LSTM, measurement matrix training

## Abstract

Video compression sensing can use a few measurements to obtain the original video by reconstruction algorithms. There is a natural correlation between video frames, and how to exploit this feature becomes the key to improving the reconstruction quality. More and more deep learning-based video compression sensing (VCS) methods are proposed. Some methods overlook interframe information, so they fail to achieve satisfactory reconstruction quality. Some use complex network structures to exploit the interframe information, but it increases the parameters and makes the training process more complicated. To overcome the limitations of existing VCS methods, we propose an efficient end-to-end VCS network, which integrates the measurement and reconstruction into one whole framework. In the measurement part, we train a measurement matrix rather than a pre-prepared random matrix, which fits the video reconstruction task better. An unfolded LSTM network is utilized in the reconstruction part, deeply fusing the intra- and interframe spatial–temporal information. The proposed method has higher reconstruction accuracy than existing video compression sensing networks and even performs well at measurement ratios as low as 0.01.

## 1. Introduction

It is a dynamic world, and the video captures this world of objects and movement. The video is composed of continuous pictures. Generally speaking, the human eye can distinguish the frame rate is 15 frames per second [1]. More than 15 frames will be considered a motion video. The ordinary camera frame rate is generally between 30 and 60 fps, which can meet most cases of content recording, while high-speed cameras need to reach 120 fps or even higher. Although high-speed video offers rich details, it comes with higher storage space and transmission bandwidth usage.

Compressive sensing (CS) [2] can acquire the measurements of the original signal at a rate lower than the Nyquist sampling rate and use an algorithm to reconstruct the original signal. This feature makes it widely used in the video field. On the one hand, it can be applied to construct high-speed cameras. By compressing consecutive frames into one frame at one measurement, it is possible to use a plain low-speed camera sensor to achieve high-speed cameras, e.g., single-pixel camera [3], single-coded exposure camera [4], and coded strobing camera [5]. On the other hand, the VCS algorithm can alleviate the enormous demand for massive storage and transmission bandwidth required for video. VCS allows transmitting the video under 100× compression, significantly improving transmission efficiency and quickly reconstructing it at the receiving end.

According to the measurement of video frames, the existing VCS methods can be divided into temporal multiplexing VCS (TVCS) and spatial multiplexing VCS (SVCS). TVCS obtains a 2D measurement frame from sampling across the temporal axis, which superimposes *k* measurement frames into one frame. Its measurement ratio is 1/k. The method proposed in [4,6,7,8,9,10] belongs to this category. These methods can obtain a high spatial resolution, which is generally implemented on sensors with low frame rates.

The SVCS is derived from single-pixel cameras [3]. It uses a programmable high-speed photodetector to measure the image. Only one measurement value is output for each measurement. A set of measurement values is obtained after multiple measurements, which is used to reconstruct the original video frame. Figure 1 shows the SVCS process. The size of the input video frame is W×H=N, and it can be vectorized as $x\in R^{N}$. The measurement matrix (MM) is expressed as $\Phi \in R^{M\times N}$, which can be viewed as a set of *M* patterns, and each row of the MM corresponds to a vectorized pattern. The measurement process is to make the inner product of each pattern and the original signal, output a single measurement value, and obtain the measurement vector $y\in R^{M}$ after *M* times calculation. The SVCS is formulated as follows:(1)y=Φx

Compared with TVCS, the SVCS can reconstruct the original frames with fewer measurements, thus achieving a relatively low measurement ratio. As a result, it causes severe loss of image information in the measurement, leading to decreased reconstruction quality. In this paper, our proposed method belongs to SVCS. Therefore, making a tradeoff between measurement ratio and reconstruction quality is one problem to be solved.

In the past ten years, many researchers have proposed VCS reconstruction algorithms based on optimization algorithms in the image field, e.g., [6,8,11,12]. Usually, these algorithms are based on the sparse prior of the image signal and use convex optimization or greedy algorithms to solve the reconstruction problem iteratively. These methods inevitably bring the problem of high computational complexity. As the resolution of the image increases, the time consumption for computation increases exponentially, making it challenging to meet real-time requirements. In recent years, with the wide application of deep learning, more and more methods based on deep learning have been proposed to solve the problem of VCS, e.g., [9,13,14,15,16,17]. These methods learn the inverse mapping directly from the measurements to the original signal through the neural networks. The reconstructed signal can be calculated through a feedforward network, which is less computationally complex than the iterative optimization algorithm. The real-time performance is improved, and the reconstruction quality is substantially improved.

The video consists of continuous frames. The video frame sequence contains the motion information of the objects in the scene. Due to the natural correlation between frames, it is possible to use spatial–temporal information, which is the key to improving the quality of reconstructed frames in the case of SVCS. In [16,18,19], the spatial–temporal information of video is fully considered when studying the VCS problem, and the spatial–temporal features of video frames are extracted using deep networks to enhance the reconstruction quality. The CSVideoNet [16] uses a classical LSTM network to extract motion features of continuous frames. In [18], Hybrid-3DNet is proposed to extract spatial information by convolving video segments using a 3D convolutional network, while VCSNet in [19] uses CNN residual connections to transfer interframe information and achieve inter-frame multi-level compensation.

This paper uses an unfolded LSTM structure to model the spatial–temporal feature in the video frame sequence. This structure is first proposed in [20] for solving sparse Bayesian learning optimization problems by mapping the traditional process of iterative SBL to an LSTM structure. The model shows good convergence performance using the unfolded LSTM for sparse optimization, which can greatly accelerate the SBL optimization solution process. Inspired by this method, we try to use the unfolded LSTM model in the VCS problem, and it provides surprisingly good performance. In the experiment, we found that the structure can not only take advantage of the LSTM’s property of long-time memory of sequences to efficiently fuse the spatial–temporal information from intra- and interframes but also fix the LSTM, which originally iterates according to the length of sequences, to a finite length. In that case, it essentially forms a feedforward network, which converges rapidly in the training process compared with the classical LSTM network. It can greatly reduce training time consumption and make the reconstruction process faster.

In addition, compared with CSVideoNet and VCSNet, since the proposed method does not adopt the multi-layer CNN structure, we use Xavier [21] to initialize the network parameters without pre-training, which makes our proposed network more efficient.

The contributions of this paper are summarized as follows:A unified end-to-end deep learning VCS reconstruction framework is proposed, including the measurement and reconstruction part, both of which consist of deep neural networks. We train the framework using a single loss function in a data-driven method, which makes the training process more efficient and faster;The spatial–temporal information fusion reconstruction with multiple sampling rates is accomplished by using the unfolded LSTM network, which obtains better reconstruction effects and improves the convergence performance significantly;Compared with the existing VCS methods, we demonstrate that the proposed network provides better reconstruction results under a wide range of measurement ratios, even under the measurement ratio as low as 0.01. We also experimentally demonstrate that the network has good convergence without pre-training and converges faster than the comparison methods.

## 2. Related Work

### 2.1. Conventional VCS Algorithm

From Eqution (Equation 1), the compressive sensing problem is reconstructing the original signal by solving an underdetermined equation, which is an NP-hard problem. There are some designed video signal priors for conventional VCS algorithms, e.g., Gaussian prior, sparse prior and TV prior, as regularization terms to the equations to constrain the possible solutions in a particular range. In [4], the Gaussian mixture model (GMM) is extended to CS territory. The method assumes that the pixels in video blocks obey the GMM distribution and use GMM to model the spatial–temporal video blocks. Video frames are reconstructed by E-M iteration. In [22], the author considers the total variational minimum can be used as the regularization. The generalized alternating projection (GAP) algorithm is used to solve the optimization problem. In [8], the author obtained better reconstruction accuracy by using low-speed image frames as side information to assist video reconstruction.

### 2.2. Deep Neural Network-Based VCS Algorithms

The DFC network in [9] is the first VCS reconstruction algorithm using a deep neural network. It uses a fully connected network to learn the direct mapping from the measurements to the original signal, solving the underdetermined problem using a data-driven approach. Compared to conventional VCS algorithms, it reduces computational complexity and improves reconstruction quality. In [23], the author demonstrates that the learned measurement matrix is superior to the random matrix. It can achieve better reconstruction results by integrating the measurement matrix into a unified training framework and learning it through network training. However, using a simple, fully connected network makes it difficult to exploit spatial–temporal information efficiently.

In order to solve the problems above, CSVideoNet [16] uses a CNN+LSTM structure to mine spatial–temporal information, enhancing the reconstruction quality. In order to balance the high compression rate and reconstruction accuracy, the video is grouped in fixed length, taking the first frame of each group as the keyframe with a low measurement ratio. The LSTM network is used to extract spatial–temporal features of the frame group to add motion estimation to the reconstruction process. CSVideoNet significantly improves the reconstruction quality. However, a pre-defined random measurement matrix in its measurement phase may not be adapted to the video task and limit performance improvement.

VCSNet [19] is a well-performing network composed entirely of a CNN structure. The training process contains multi-stages, with the measurement matrix being trained first, followed by the reconstruction part. The reconstruction part is further divided into two stages: keyframe training and non-keyframe training. A multi-level feature compensation method is used in these two parts, which compensates for the reconstruction of non-key frames by using high-quality keyframes. Using CNN networks with multi-level compensation can only compensate for the loss of spatial information in non-key frames. In contrast, the temporal information contained in the video is hard to be learned. In addition, up to 2K + 2 loss functions are used in the training process, and such a multi-loss network is difficult to train.

## 3. Video Compressive Sensing Reconstruction Using Unfolded LSTM

### 3.1. Overview of the Proposed Framework

This section will give the details of the proposed end-to-end VCS reconstruction network. The network structure is shown in Figure 2. It can be divided into encoding and decoding parts.

The encoding part provides a learned measurement matrix, which is equivalent to a CNN’s convolution process. Since the proposed network belongs to SVCS and will lose more spatial resolution in the measurement process due to the low measurement ratio, the keyframe technique is introduced. The input video is split into multiple groups of pictures (GOPs), with the first frame of each GOP as the keyframe and the rest as non-key frames. A higher measurement ratio is applied to the keyframe to retain more spatial information. In comparison, a lower measurement ratio for non-key frames reduces the overall sampling rate of the GOP.

The decoding part includes the initial reconstruction and the deep spatial–temporal information fusion reconstruction. Initial reconstruction is used in many VCS methods, allowing a smoother transition to the subsequent enhanced reconstruction part. In the first stage of the decoding part, there is another CNN to perform an inverse convolution on the measurements to obtain an initial reconstructed frame, which has the same dimensions as the original frame. The proxy keyframes and non-key frames are input into the next stage, the unfolded LSTM network, for the dynamic information fusion reconstruction. The network can preserve the motion information in frame sequence and fully extract the spatial–temporal features, achieving high-fidelity reconstructed video.

For the training, we use a single mean square error (MSE) as the loss function, directly comparing the difference between the reconstructed and the original video segments. It makes the training process simple and efficient, avoiding the problems of parameter balance and poor convergence in multi-loss training.

### 3.2. Encoding Part

In many VCS algorithms, the measurement matrix uses a pre-defined random matrix for measurement. However, the study in [23] indicates that the measurement matrix learned from data has better performance in the reconstruction. The details of the encoding part are shown in Figure 3.

Each video frame size W×H can be divided into $w_{p}\times h_{p}=P$ image blocks of size without overlapping. We denote the $i^{th}$ patch as $x_{i}=\left[x_{i}^{1},x_{i}^{2},\cdots ,x_{i}^{M}\right]$, where M=s×s denotes the pixels of each small block. As illustrated in Figure 2, for the $i^{th}$ patch, the measurement process can be formulated as:(2)$$y_{i}=\Phi x_{i}$$
where $\Phi ={\left[\phi_{1},\cdots ,\phi_{k}\right]}^{T}$, $\phi_{j}$ denotes the column vector, which is vectorized from the $j^{th}$ pattern. $y_{i}=\left[y_{i}^{1},\cdots ,y_{i}^{k}\right]$, $y_{i}^{j}$ is the inner product of $\phi_{j}$ and $x_{i}$, which denotes as $y_{i}^{j}=\phi_{j}^{T}•x_{i}$. The measurement process for each patch can be equated to convolution operation by CNN, which has *k* kernels of size s×s with step length *s*. We arranged the format in order (number of image channels, image height, image width); the size of the input is denoted as (1,H,W) and output is denoted as $\left(k,h_{p},w_{p}\right)$.

For a GOP of length *T*, the process from each video frame $X_{i}=\left[x_{1},\cdots ,x_{P}\right]$ to the measurements $Y_{i}=\left[y_{1},\cdots ,y_{P}\right]$ can be formulated as:(3)$$Y_{i}=\Phi X_{i}$$
where $Y_{i}$ is of size $\left(k,h_{p},w_{p}\right)$ and $X_{i}$ is of size H×W; thus, the measurement ratio (MR) is calculated as:(4)$$MR=\frac{k\times h_{p}\times w_{p}}{H\times W}=\frac{k\times \frac{H}{s}\times \frac{W}{s}}{H\times W}=\frac{k}{s\times s}$$

From Eqution (Equation 4), it can be concluded that the size of the measurements can be easily controlled by the number of convolution kernels *k*. For each GOP containing *T* frames, denoted $k_{1},k_{2}$ be the number of convolutional kernels of the keyframe and non-keyframe, respectively; then, the global measurement ratio can be calculated as:(5)$$MR_{gop}=\frac{k_{1}+k_{2}\times \left(T-1\right)}{s\times s\times T}$$

### 3.3. Decoding Part

The decoding part can be divided into the initial reconstruction and the deep spatial–temporal information fusion reconstruction.

#### 3.3.1. Initial Reconstruction

The initial reconstruction is used in many VCS algorithms. Firstly, a simple inverse transformation of the measurements is completed to obtain a proxy frame with the same dimensions as the target frame and then further refine the proxy frame in the next stage. Figure 4 shows the details. The initial reconstruction process can be seen as the inverse of the encoding part, converting measurements $Y_{i}$ with dim $k\times h_{p}\times w_{p}$ to proxy frame ${\tilde{X}}_{i}$ with dim 1×H×W. Here, we focus on the inverse process. For each patch $x_{i}$, we can also use a CNN to obtain the proxy frame; just set the kernel of size (s×s)×1×1 and step length to 1. By convolving the measurements $y_{i}$, we obtain a column vector ${\tilde{x}}_{i}$ with dimension 1×1×(s×s). Reshaping ${\tilde{x}}_{i}$ to 1×s×s, we obtain the initial reconstruction of the patch. Extending the convolution operation to the whole frame, we can obtain the convolutional output of size $\left(s\times s\right)\times h_{p}\times w_{p}$. After reshaping, the proxy frame of size 1×H×W is finally recovered.

#### 3.3.2. Deep Spatial–Temporal Information Fusion

For a GOP of length *T*, which has undergone a measurement and initial reconstruction phase, we put one proxy keyframe ${\tilde{X}}_{key}$ and (T−1) proxy non-keyframes ${\tilde{X}}_{\;\mathrm{nonkey}\;}^{i}\left(i=1,2,\cdots ,T-1\right)$ together in sequential order and denote them as ${\tilde{X}}_{GOP}$. For continuous video frames, the interframes contain part of the scene invariance and the motion information of the objects. To fully use the spatial–temporal information, we propose a spatial–temporal information fusion module based on unfolded LSTM, which helps aggregate motion and spatial features for each frame.

Figure 5 shows the full details of the network. The unfolded LSTM network structure consists of LSTM cells. The update process is as follows:(6)$$\begin{array}{cc} c_{j}^{\left(t\right)} & =f_{j}^{\left(t\right)}⊙c_{j}^{\left(t-1\right)}+i_{j}^{\left(t\right)}⊙{\tilde{c}}_{j}^{\left(t-1\right)}\\ h_{j}^{\left(t\right)} & =o_{j}^{\left(t\right)}⊙\mathrm{Tanh}\left(c_{j}^{\left(t\right)}\right)\\ i_{j}^{\left(t\right)} & =\sigma (W_{i_{j}}\left[h_{j}^{\left(t-1\right)},I_{j}^{\left(t\right)}\right])\\ f_{j}^{\left(t\right)} & =\sigma (W_{f_{j}}\left[h_{j}^{\left(t-1\right)},I_{j}^{\left(t\right)}\right])\\ o_{j}^{\left(t\right)} & =\sigma (W_{o_{j}}\left[h_{j}^{\left(t-1\right)},I_{j}^{\left(t\right)}\right])\\ {\tilde{c}}_{j}^{\left(t\right)} & =\tanh(W_{c_{j}}\left[h_{j}^{\left(t-1\right)},I_{j}^{\left(t\right)}\right]) \end{array}$$
where the subscript *j* indicates the index of the stacked layers of the LSTM, $I_{j}^{\left(t\right)}$ denotes the input of the first LSTM cell, $I_{1}^{\left(t\right)}=X_{GOP}^{\left(t\right)}$, $I_{j}^{\left(t\right)}=h_{j-1}^{\left(t\right)}\left(j>1\right)$. $h_{j}^{\left(t\right)}$ and $c_{j}^{\left(t\right)}$ denote, respectively, the hidden state and memory cell of the LSTM cell of the $j^{th}$ layer at the *t* moment. $i_{j}^{\left(t\right)}$, $f_{j}^{\left(t\right)}$, $o_{j}^{\left(t\right)}$ correspond to the input gate, forget gate and output gate. We stack *r* layers of the LSTM cell to construct the LSTM stack module, denoted as $F_{LSTM-Stacked}$, and the process of update can be formulated as:(7)$$\begin{array}{cc} & =F_{LSTM-\;\mathrm{Stacked}\;}\left(H^{\left(t-1\right)},{\tilde{X}}_{GOP}^{\left(t\right)};\theta \right)\\ q^{\left(t\right)} & =h_{r}^{\left(t\right)} \end{array}$$
where $H^{\left(t-1\right)}=\left[h_{1}^{\left(t-1\right)},\cdots ,h_{r}^{\left(t-1\right)}\right]$ contains all the hidden states of each LSTM cell in the stacking module, θ denotes the network parameters, and $q^{\left(t\right)}$ indicates the output of the stacking module, which is the hidden state of the last LSTM cell in the stack.

Unfolding the LSTM stack in *l* steps, we can obtain a feedforward network. During the forward stage, the input ${\tilde{X}}_{GOP}^{\left(t\right)}$ is broadcast to the lowest RNN cell at each unrolled step. The input of the hidden state of the unfolding step comes from the hidden state of the $d^{th}$ unfolding step. The output of ${\left(d-1\right)}^{th}$ the last unfolding step is passed through a fully connected network to obtain the final reconstructed video ${\hat{X}}_{GOP}$. Denoting the $d^{th}$ unfolding step as $F_{LSTM-Stacked}^{d}$ and fully connected network as $F_{FC}\left(•\right)$, the process can be formulated as:(8)$$\begin{array}{cc} & =F_{LSTM-Stacked}^{d}\left(H_{d-1}^{\left(t\right)},{\tilde{X}}_{GOP}^{\left(t\right)};\theta \right)\\ {\hat{X}}_{GOP} & =F_{FC}\left(q^{\left(\;\mathrm{all}\;\right)}\right)\\ q^{\left(\;\mathrm{all}\;\right)} & =[q^{\left(1\right)},q^{\left(2\right)},\cdots ,q^{\left(T\right)}] \end{array}$$

### 3.4. Training

The VCS algorithm proposed is an end-to-end network architecture. The mean square error (MSE) is used as the loss function to train the whole network, which is defined as:(9)$$\mathrm{LOSS}=\frac{1}{T}\sum_{i}^{T}{∥{\hat{X}}_{GOP}^{i}-X_{GOP}^{i}∥}_{2}^{2}$$

Adam [24] optimizer is used for training to optimize the network parameters. Compared to the multi-stage training strategy VCSNet, the whole network is trained under a unified framework with only one global loss function, making the whole network training process concise and efficient. We use Peak Signal-to-Noise Ratio (PSNR) and Structure Similarity (SSIM) [25] as the evaluation index of image reconstruction quality. PSNR, as a widely used metric for image reconstruction quality in engineering, is calculated based on the error between the corresponding pixels. The larger the value, the smaller the image distortion, which can reflect the final video reconstruction quality to a certain extent. SSIM is an evaluation metric based on the structural information of the scene, comparing the brightness, contrast and structural information in the image block, which to a certain extent can reflect the human eye’s perception of picture similarity. The reconstruction quality can be better evaluated with these two metrics.

## 4. Experiment and Analysis

### 4.1. Dataset

UCF101 [26] is used as the dataset to benchmark our proposed network. The dataset collects and organizes 101 kinds of human action videos and contains up to 13,000 video clips with a duration of more than 27 h. Videos in the dataset have a resolution of 320×240 and are sampled at 25 fps. We selected 20 types of action videos as the training set and five types of entirely different action videos as the validation set. For comparison with the existing VCS algorithms, we crop the video frame into two different sizes of 96×96, 160×160 with the center part and split them into GOPs of length 10. We obtain 271,930 and 38,000 GOPs for training and testing, respectively.

### 4.2. Implementation Details

We set the keyframe measurement ratio $MR_{key}$ to 0.5, 0.2, 0.1 for the different comparison experiments. According to Eqution (Equation 5), the corresponding non-key frame ratio $MR_{nonkey}$ can be calculated based on the given $MR_{GOP}$. We set s=32 and $k_{1},k_{2}$ can be calculated according to Eqution (Equation 4). Our network is trained for 400 epochs with a batch size of 200. The network parameters are initialized using Xavier [21], and no pre-training is required. Adam is applied as the optimizer, and the learning rate is set to 0.001. Our network is trained for 400 epochs with a mini-batch size of 200.

### 4.3. Compared with Existing VCS Algorithms

In order to fully validate the effectiveness of the proposed method, we compare the proposed method with the existing VCS algorithms based on a deep neural network, including CSVideoNet [16], VCSNet [19], DFC [9], and C2B [27]. PSNR and SSIM are applied for performance evaluation.

In order to objectively compare the performance of the proposed algorithm with the existing methods, we set the same experimental condition as the comparison algorithms and compare the reconstruction quality separately. Since CSVideoNet and VCSNet all belong to SVCS, the proposed algorithm is compared with these two first.

The comparison with VCSNet is shown in Table 1. We use the parameter settings of VCSNet. Since keyframe MR is set fixed to 0.5, the length of GOP is 8 and non-key frame MR is set to 0.1, 0.01; the corresponding global MR of GOPs is 0.15, 0.07. The reconstruction quality of the proposed method is 2.52 db and 3.08 db higher than that of VCSNet under two different ratios, respectively.

The comparison with CSVideoNet is shown in Table 2. The experiment of CSVideoNet challenges the ultimate performance of VCS by setting three very low global MR values of 0.04, 0.02 and 0.01. We set the same experimental parameters, and it can be seen that the proposed method outperforms CSVideoNet by nearly 5 db on average in the three comparison groups.

The VCS methods belonging to TVCS usually set the experiment to compress 16 original video frames into one measurement frame, where MR=1/16 (0.0625). We compare the proposed algorithm with the existing algorithms under MR=1/16, so we use $MR_{key}=0.2$, $MR_{nonkey}=0.047$, and correspondingly, $k_{1}=51$, $k_{2}=12$. The results are shown in Table 3, and our proposed method achieves the best performance. We also give some reconstructed frames as a visual comparison in Figure 6.

Obviously, through the above comparison experiments, we can conclude that the proposed algorithm performs better than the existing VCS methods under various MRs.

### 4.4. The Effectiveness of the Unfolded LSTM Structure

To demonstrate the effectiveness of this unfolded LSTM structure in the algorithm, we design comparative experiments where unfolded LSTM in the proposed algorithm is replaced with a classic LSTM structure. The experimental results are shown in Table 4. It can be seen that the average reconstruction accuracy is higher with the unfolded LSTM approach than with the classic LSTM, which indicates that the unfolded LSTM structure can improve the reconstruction quality better than the classic LSTM structure.

### 4.5. Optimal Measurement Ratio Allocation for Keyframes and Non-Keyframes

We also conduct additional experiments to explore the measurement ratio allocation for keyframes and non-keyframes at the fixed $MR_{GOP}$. The experimental results are shown in Table 5. $MR_{GOP}$ is set to 1/16. We found that appropriately reducing the measurement ratio of keyframes and increasing the measurement ratio of non-keyframes can improve the reconstruction results. We speculate that this phenomenon is due to a large amount of redundancy in the video frame information. A relatively low measurement ratio may exist for the keyframe to retain most of the spatial information.

### 4.6. Convergence Performance and Time Complexity

We designed experiments to compare the convergence performance of different methods. The same experimental conditions are set for each method, and then, we obtain the experimental results shown in Figure 7. The curves given are the variation of the reconstruction accuracy of the network on the validation set in each training epoch. From this figure, we can see that the reconstruction accuracy of CSVideoNet and VCSNet on the validation is are not higher than 20 db, indicating that the networks are easily trapped in local minima and challenging to converge to the optimum. The proposed method in this paper can converge to the appropriate interval after one epoch and achieves an accuracy of more than 30 db on the test set. Table 6 shows the time spent for each training epoch of the network during the training process. It takes much less time for the proposed network to train than CSVideoNet and VCSNet. From the above, the proposed network not only has better convergence performance in the training process but also has lower time complexity and higher overall efficiency.

**Figure 6 sensors-22-07172-f006:**
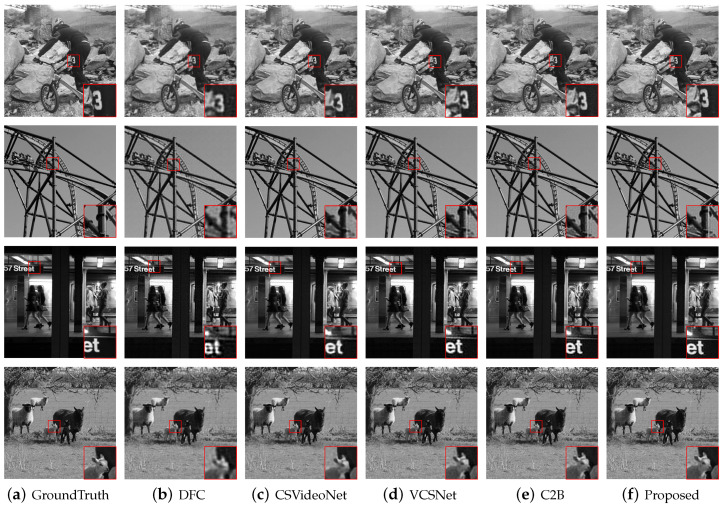
The comparison of reconstructed frames between the proposed and DFC, CSVideo, VCSNet and C2B methods.

**Figure 7 sensors-22-07172-f007:**
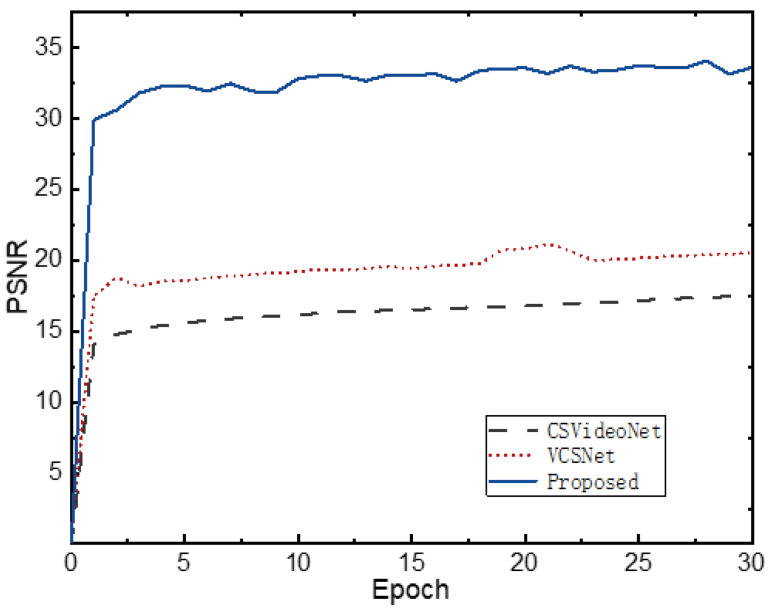
Variation curve of model reconstruction performance with the number of training epochs.

## 5. Conclusions

In this paper, we propose a novel unified end-to-end VCS reconstruction network. In the measurement sampling stage, the CNN network integrates the measurement process into the network and learns a better measurement matrix. In the reconstruction stage, the unfolded LSTM network is applied to fuse the spatial–temporal feature at the frames with different measurement ratios. Compared with the existing deep neural network-based VCS methods, the proposed network can achieve better reconstruction accuracy under the same conditions and even obtain satisfactory reconstruction results at compression rates as low as 0.02 and 0.01. In future work, we will build on this and continue exploring computer vision downstream tasks such as target tracking, re-identification, etc.

## Figures and Tables

**Figure 1 sensors-22-07172-f001:**
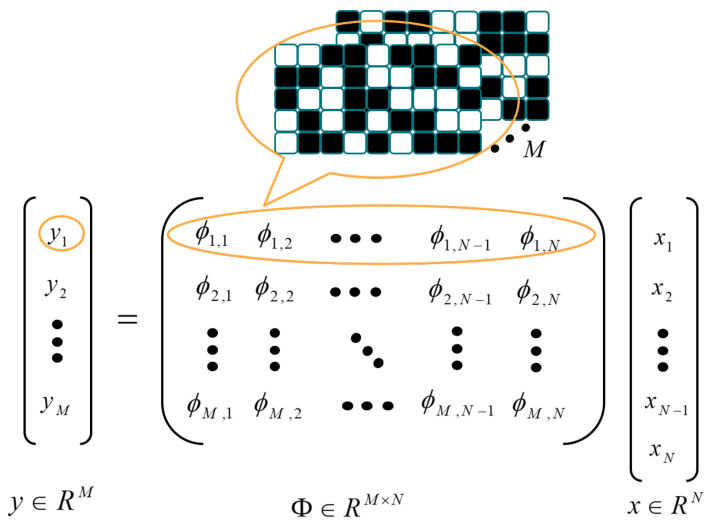
Principle of SVCS measurement process.

**Figure 2 sensors-22-07172-f002:**
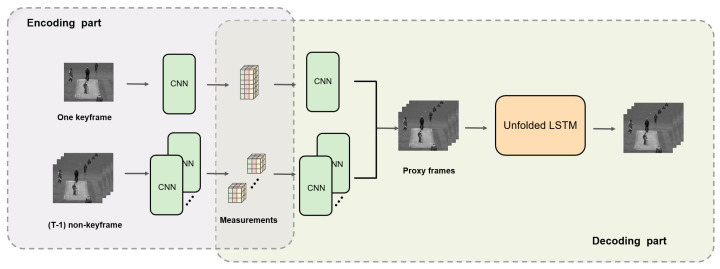
The architecture of the proposed framework.

**Figure 3 sensors-22-07172-f003:**
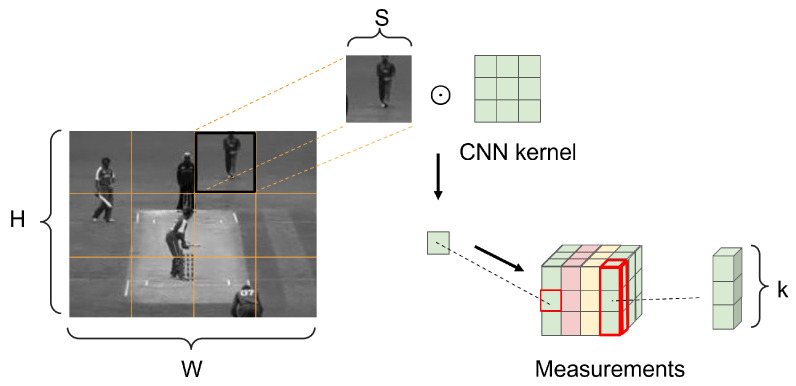
The measurement process based on CNN structure.

**Figure 4 sensors-22-07172-f004:**
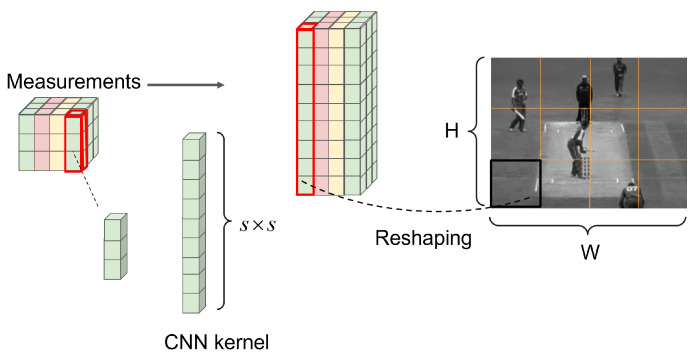
The process of the initial reconstruction.

**Figure 5 sensors-22-07172-f005:**
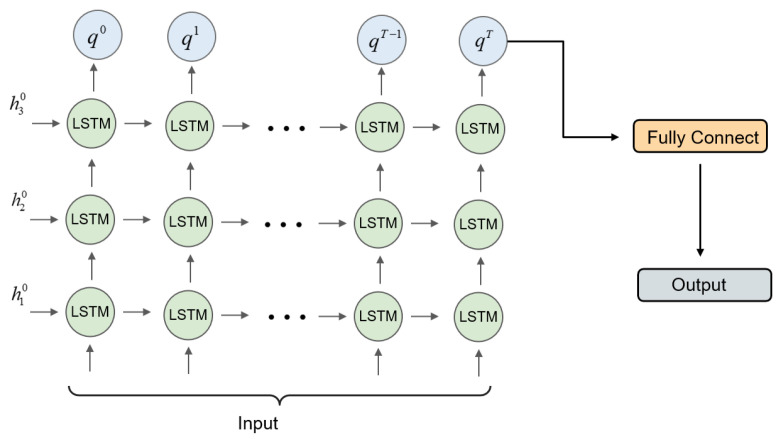
The process of the unfolded LSTM.

**Table 1 sensors-22-07172-t001:** Comparison of the proposed method with VCSNet.

Name	Ratio	Frame Size	GOP	${\mathrm{MR}}_{\mathrm{key}}$	${\mathrm{MR}}_{\mathrm{nonkey}}$	PSNR	SSIM
VCSNet	0.15	96 × 96	8	0.5	0.1	34.29	0.90
Proposed		96 × 96	8	0.5	0.1	**36.91**	**0.96**
VCSNet	0.07	96 × 96	8	0.5	0.01	29.58	0.82
Proposed		96 × 96	8	0.5	0.01	**32.66**	**0.91**

**Table 2 sensors-22-07172-t002:** Comparison of the proposed method with CSVideoNet.

Name	Ratio	Frame Size	GOP	${\mathrm{MR}}_{\mathrm{key}}$	${\mathrm{MR}}_{\mathrm{nonkey}}$	PSNR	SSIM
CSVideoNet	0.04	160 × 160	10	0.2	0.022	26.87	0.81
Proposed		160 × 160	10	0.2	0.022	**32.64**	**0.88**
CSVideoNet	0.02	160 × 160	10	0.1	0.011	25.09	0.77
Proposed		160 × 160	10	0.1	0.011	**31.11**	**0.84**
CSVideoNet	0.01	160 × 160	10	0.06	0.004	24.23	0.74
Proposed		160 × 160	10	0.06	0.004	**28.64**	**0.81**

**Table 3 sensors-22-07172-t003:** Comparison of the proposed method with TVCS methods under ratio 1/16 (0.0625).

Name	Ratio	Frame Size	GOP	${\mathrm{MR}}_{\mathrm{key}}$	${\mathrm{MR}}_{\mathrm{nonkey}}$	PSNR	SSIM
DCF	1/16	160 × 160	16	-	-	24.67	0.71
C2B		160 × 160	16	-	-	32.23	0.93
CSVideoNet		160 × 160	10	0.2	0.022	28.08	0.84
VCSNet		160 × 160	10	0.2	0.022	28.57	0.86
Proposed		160 × 160	10	0.2	0.022	**35.02**	**0.95**

**Table 4 sensors-22-07172-t004:** Comparison of the proposed method with classical LSTM structure.

Name	Ratio	Frame Size	GOP	${\mathrm{MR}}_{\mathrm{key}}$	${\mathrm{MR}}_{\mathrm{nonkey}}$	PSNR	SSIM
LSTM	0.04	160 × 160	10	0.2	0.047	34.11	0.91
Proposed		160 × 160	10	0.2	0.047	**35.02**	**0.95**
LSTM	0.02	160 × 160	10	0.2	0.022	31.50	0.81
Proposed		160 × 160	10	0.2	0.022	**32.64**	**0.91**
LSTM	0.01	160 × 160	10	0.1	0.011	30.32	0.77
Proposed		160 × 160	10	0.1	0.011	**31.11**	**0.88**

**Table 5 sensors-22-07172-t005:** Comparison of the proposed method performance in different $MR_{key}$ under the same ratio.

Frame Size	Patch Size	GOP	${\mathrm{MR}}_{\mathrm{key}}$	${\mathrm{MR}}_{\mathrm{nonkey}}$	$k_{1}$	$k_{2}$	PSNR	SSIM
96 × 96	32 × 32	10	0.5	0.014	512	14	31.88	0.81
96 × 96	32 × 32	10	0.4	0.025	409	25	33.59	0.83
96 × 96	32 × 32	10	0.3	0.036	307	37	34.58	0.86
96 × 96	32 × 32	10	0.2	0.047	204	48	**35.02**	**0.91**

**Table 6 sensors-22-07172-t006:** Comparison of the time complexity of the existing models and the proposed model.

Model	CSVideoNet	VCSNet	Proposed
Time(s)	587.70	359.61	**194.53**

## Data Availability

The database used in this article is UCF101. For details, please refer to [26].

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
