# Peer review of "Video Compressive Sensing Reconstruction Using Unfolded LSTM"

_sensors, 2022, doi:10.3390/s22197172_

Round 1
Reviewer 1 Report
The authors propose a new algorithm for video compression and reconstruction based on an unfolded LSTM network.
The paper is clearly written, the proposed algorithm and the performed computations are well explained.
The weak point of the paper is the comparison of the proposed algorithm with the previously known methods. The authors use two metrics for video reconstruction quality assessment: Peak Signal-to-Noise Ratio (PSNR) and Structure Similarity (SSIM). However, these both are image metrics and the authors do not explain how they obtained the PSNR and SSIM values given in the Tables 1-5. Are these values calculated for single images? Are these average values over all the images contained in the video, over the images in the GOP or over other defined image series? In each of these cases, a comparison based only on the average values is not strong enough to demonstrate the outperformance of the proposed algorithm. Statistical parameters such as min and max image metric values, standard deviation, or alternatively the statistical distributions (histograms), are also relevant for the comparison and should be included in the study.
The authors should also show some examples of reconstructed images (pristine image and image reconstructions using different algorithms).
A small correction is necessary in Table 6; VCSNet appears two times.
Reviewer 2 Report
The paper describe an efficient algorithm for video compressive sensing reconstruction, using an approach based on LSTM network. The authors describe the past works in the field and propose a new method. The methods are described sufficiently well and the results, which compare the reconstruction efficiency of their method with the old approaches, are clearly presented. Overall the reader can see that a lot of work have been done, and it is clearly presented.
The only problem is with the english, which is not always clear and shoud be revised.
